# Preclinical assessment of splicing modulation therapy for *ABCA4* variant c.768G>T in Stargardt disease
Dyah W. Karjosukarso [1], Femke Bukkems[1,2], Lonneke Duijkers[1], Tomasz Z. Tomkiewicz[1],
Julia Kiefmann[1], Andrei Sarlea [3], Sander Bervoets [4], Irene Vázquez-Domínguez [1], Laurie L. Molday[5],
Robert S. Molday[5], Mihai G. Netea [3], Carel B. Hoyng[2,6], Alejandro Garanto[1,7] & Rob W. J. Collin [1,2] ✉

## Abstract

**Background** Stargardt disease type 1 (STGD1) is a progressive retinal disorder caused by bi-allelic variants in the *ABCA4* gene. A recurrent variant at the exon-intron junction of exon 6, c.768G>T, causes a 35-nt elongation of exon 6 that leads to premature termination of protein synthesis.

**Methods** To correct this aberrant splicing, twenty-five 2′-*O*-methoxyethyl antisense oligonucleotides (AONs) were designed, spanning the entire exon elongation.

**Results** Testing of these AONs in patient-derived photoreceptor precursor cells and retinal organoids allow the selection of a lead candidate AON (A7 21-mer) that rescues on average 52% and 50% expression of wild-type *ABCA4* transcript and protein, respectively. In situ hybridization and probe-based ELISA demonstrate its distribution and stability in vitro and in vivo. No major safety concerns regarding off-targets, immunostimulation and toxicity are observed in transcriptomics analysis, cytokine stimulation assays in human primary immune cells, and cytotoxicity assays.

**Conclusions** Additional optimization and in vivo studies will be performed to further investigate the lead candidate. Considering the high prevalence of this variant, a substantial number of patients are likely to benefit from a successful further development and implementation of this therapy.

## Plain language summary

Stargardt disease is an inherited blindness that is caused by defect(s) in the *ABCA4* gene. It starts with central vision loss which may gradually lead to complete blindness. We designed drugs to modify a common *ABCA4* gene mutation and restore the correct function of the ABCA4 protein. We evaluate these drugs in cell and mouse models of eye disease to select the most promising one. Further studies could result in the start of testing the drug in patients with Stargardt disease.

Stargardt disease type 1 (STGD1, OMIM 248200) is an inherited retinal disease that occurs in approximately 1:10,000 people[1]. It is a progressive condition that starts with central vision loss which may gradually lead to complete blindness. Although the early signs of STGD1 are typically observed during adolescence and young adulthood, the age of onset can vary greatly, depending on the severity of the causative mutations[2]. Genetically, bi-allelic variants in *ABCA4* are the underlying cause of this recessive disease[3,4].

*ABCA4* encodes the ATP Binding Cassette Subfamily A Member 4 (ABCA4), a transmembrane protein that is mainly expressed in photoreceptor cells of the retina[5]. As an ABC-transporter, it facilitates the removal of toxic by-products of the visual cycle, such as N-retinylidene-N-retinlyethanolamine. Dysfunction of ABCA4 leads to the accumulation of this toxic compound in photoreceptors and the retinal pigment epithelium, ultimately leading to the degeneration of these cells[6].

To date, more than 2400 different *ABCA4* variants have been described (www.lovd.nl/ABCA4). Variants in *ABCA4* are distributed over the entire gene, both in the coding and non-coding regions. The vast majority of variants in the non-coding regions are near exon or deep-intronic variants that lead to abnormal splicing[2]. A noncanonical splice site variant,

[1]Department of Human Genetics, Research Institute for Medical Innovation, Radboud University Medical Center, Nijmegen, The Netherlands. [2]Astherna B.V., Nijmegen, The Netherlands. [3]Department of Internal Medicine and Radboud Center for Infectious Diseases, Research Institute for Medical Innovation, Radboud University Medical Centre, Nijmegen, The Netherlands. [4]Radboudumc Technology Center Bioinformatics, Research Institute for Medical Innovation, Radboud University Medical Center, Nijmegen, The Netherlands. [5]Department of Biochemistry and Molecular Biology, The University of British Columbia, Vancouver, Canada. [6]Department of Ophthalmology, Research Institute for Medical Innovation, Radboud University Medical Center, Nijmegen, The Netherlands. [7]Department of Pediatrics, Amalia Children's Hospital, Research Institute for Medical Innovation, Radboud University Medical Center, Nijmegen, The Netherlands. ✉e-mail: rob.collin@radboudumc.nl

c.768G>T, is a recurrent variant calculated to be present in a few thousand cases with STGD1. Located at the last nucleotide of *ABCA4* exon 6, this synonymous variant reduces the strength of the original splice leading to a 35-nt elongation of exon 6, hence causing a frameshift and loss-of-function[7].

The high prevalence of c.768G>T renders it an interesting therapeutic target. Twenty-five 2'-*O*-methoxyethyl antisense oligonucleotides (AONs) with phosphorothioate linkage (2'MOE/PS) were designed to rescue the splicing defect caused by this variant. Screening of these AONs in photoreceptor precursor cells, allowed the selection of top 10 AONs based on the rescue efficacy. A follow-up test of these 10 AONs in retinal organoids (ROs) revealed a lead candidate. The outcome of investigation at the protein level corroborates the efficacy of this AON to rescue ABCA4 expression. Potential off-targets, stability, immunostimulatory profile and cytotoxicity of the therapeutic molecule were thoroughly assessed and revealed no concerns. Taken together, these results demonstrate that the lead candidate AON has great potential to correct the aberrant splicing caused by *ABCA4* c.768G>T, allowing further investigation of this approach towards clinical implementation.

## Materials and Methods
### Ethical declarations
Written informed consent was obtained from all participants. The use of patient-derived iPSCs and fibroblasts was approved by the Local Ethics Committee (Centrale Commisie Mensgebonden Onderzoek, CMO-light, protocol number 2015-1543) and the use of PBMC from healthy donors is registered under NL84281.091.23. This study was conducted in adherence to the tenets of Declaration of Helsinki. The mouse experiments were approved by the local Animal Experimentation Committee (Centrale Commissie Dierproeven, RU-DEC-2016-0050, work protocol 2016-0050-022 and 2016-0050-024) and were performed according to the Dutch law (Wet op Dierproeven 1996).

### Cell culture
Fibroblasts from a patient homozygously carrying the c.768G>T *ABCA4* variant and those from a control individual, HEK293T (Human embryonic kidney 293T, ATCC CRL-3216), and hTERT RPE-1 (human telomerase reverse transcriptase immortalized retinal pigment epithelial, ATCC CRL-4000) were cultured in corresponding culture media detailed in Supplementary Table 1. These cells were cultured at 37 °C 5% $CO_2$ and split twice a week.

Patient-derived induced pluripotent stem cells (iPSCs) included in this study homozygously carried the c.768G>T variant[8], which was corrected in the isogenic control line[9]. The iPSCs were cultured in Essential 8 flex medium (Gibco) on Geltrex coated culture dishes (1:100 dilution, Gibco) at 37 °C 5% $CO_2$. Colonies were split every 5-7 days.

### Differentiation towards photoreceptor precursor cells (PPCs) and retinal organoids (ROs)
The differentiation protocol that was used is a slightly modified version of that was previously described[10]. The differentiation was started by culturing confluent iPSCs culture in Essential 6 medium (Gibco) for 2 days. Neural induction was initiated at day 2 of the differentiation by introducing neural induction medium. At day 6, BMP4 (Sigma Aldrich) was introduced at 1.5 nM final concentration, which is gradually decreased by refreshing half of the medium every other day. For PPCs, this was performed until the cells were harvested at day 30. For ROs, neuroretinal vesicles were collected between day 20-40 of the differentiation. Once collected in 96-well ultra-low attachment U-bottom plates (faCellitate), the 3D structures were given retinal differentiation medium twice. Subsequently, the medium was changed to retinal maturation medium 1. At day 50, 9-*cis* retinal (Sigma Aldrich) at a final concentration of 1 µM was added to the medium. The medium was changed to retinal maturation medium 2 supplemented with 0.5 µM 9-*cis* retinal after 70 days of differentiation. At day 100, the 3D structures were visually examined under Evos XL Core microscope for lamination as a RO indicator. The laminated organoids were transferred

individually to 25-well low attachment plates (Sterilin, Thermo Fisher Scientific) and subjected to retinal maturation medium 3. Half medium refreshments were performed throughout the 3D culture phase, three times a week for the 96-well plate format and twice a week for the 25-well plate format. The details of media composition are provided in Supplementary Table 1. Mature ROs were used for experiments between days 180 and 250.

### Characterization of PPCs and ROs
PPCs were characterized by measuring the expression level of marker genes such using quantitative polymerase chain reaction (qPCR). Briefly, RNA was isolated and cDNA was synthesized to be used as template. The details of this procedure are described in the corresponding section. Quantitative PCR was performed using GoTaq qPCR master mix (Promega) and Quantstudio 3 (Applied Biosystems). The primers used are listed in Supplementary Table 2.

ROs were characterized by immunohistochemistry using antibodies against rod marker rhodopsin (RHO), cone marker arrestin (ARR3) and the photoreceptor marker CRX. The antibodies used are listed in Supplementary Table 3 and the details of the immunohistochemistry procedure are described in the corresponding section. The expression of marker genes were also extracted from FPKM-normalized (fragments per kilobase of transcript per million mapped reads) transcriptome data.

### AON delivery to PPCs and ROs
Twenty-five 2'-*O*-methoxyethyl AONs with phosphorothioate linkage (2'MOE/PS) were designed covering the aberrant exon elongation. The AON sequences used in this study are listed in Supplementary Table 4 and were custom synthesized by Kaneka Eurogentec S.A (Belgium, www.eurogentec.www.eurogentec.com/en/). For AON screening in PPCs, 5 µM of AON was delivered gymnotically at day 20 of differentiation. For AON testing in ROs, gymnotic delivery of the AONs was performed when the ROs reached at least day 180 of the differentiation in a final concentration of 10 µM unless otherwise indicated. Typically 1 RO per condition was used for experiments with RT-PCR or imaging read-out, whereas a pool of 3-4 ROs were required for Western blot and RNA-seq experiments. In case more than one RO was needed, individual ROs were treated with AON and pooled at harvest. The treatment duration ranged from 10 to 60 d and are indicated for each experiment presented. All AON deliveries were single delivery, including for the long-term experiments. Half-medium refreshment was carried out three times a week during each experiment. For transcript analysis, PPCs or ROs were treated with 0.1 mg/ml cycloheximide (Sigma Aldrich) 6 h prior to harvest in order to inhibit nonsense-mediated decay. PPCs were harvested by adding RA1 buffer from Nucleospin RNA Mini kit (Macherey Nagel) to the cells followed by snap-freezing the collected lysate, whereas ROs were harvested by snap-freezing, except for immunostaining which fixation method is described in another section.

### RNA isolation and Reverse transcriptase-polymerase chain reaction (RT-PCR)
RNA isolation was performed using Nucleospin RNA Mini kit (Macherey Nagel) following manufacturer's instructions. RO was submerged in RA1 buffer provided by the kit and homogenized using glass beads and TissueLyserIII (Qiagen) at the frequency of 30/sec for 1 min. Reverse transcription was carried out with iScript cDNA synthesis kit (Bio-Rad). From the PPCs, 200 ng and 50 ng cDNA were used as PCR template for *ABCA4* and *ACTB* transcript, respectively. Due to the higher expression of *ABCA4* in ROs, less template was needed, namely 25 ng for *ABCA4* and 10 ng for *ACTB*. The PCR mix contains 1x PCR buffer (Roche), 0.2 mM dNTP (Thermo Fisher Scientific), 0.2 µM forward/reverse primers (Supplementary Table 2), 0.5 U Taq (Roche) and 10% Q solution (Qiagen). An annealing temperature of 58 °C and 30 s elongation at 72 °C were employed for 35 cycles. Semi-quantification of the bands observed was performed with Fiji 1.53. The transcripts were empirically quantified with Agilent Tapestation High Sensitivity D1000. The %WT transcript level was calculated based on the

ratio of aberrant, correct, and partial exon 6 deletion transcripts in each sample, with the total signal set at 100%.

## Western blot

ROs were incubated in modified radioimmunoprecipitation assay (RIPA) buffer (50 mM Tris-HCl pH 7.5, 150 mM NaCl, 1% NP-40, 0.75% SDS, 0.5% sodium deoxycholate, 1 mM Ethylenediaminetetraacetic acid (EDTA)) supplemented with 1× Complete Protease Inhibitor Cocktail (Roche) for 1 h at 4 °C followed by homogenization with glass beads and TissueLyserIII (Qiagen) at the frequency of 30/s for 1 min. The obtained lysates were sonicated followed by total protein quantification using Pierce BCA Protein Assay kit (Thermo Fisher Scientific). Afterwards, 30 µg total protein was loaded onto Mini-PROTEAN TGX Stain-Free 4-15% gel (Bio-Rad) followed by transfer onto nitrocellulose membrane (TransBlot Turbo Transfer Pack, Bio-Rad). The membranes were blocked for 1 h at room temperature (RT) with 5% non-fat dry milk (Santa Cruz) and incubated overnight at 4 °C with primary antibodies (Supplementary Table 3) diluted 1:1000 in blocking buffer. Unbound antibodies were washed 3 × 5 min with phosphate buffered saline (PBS) + 0.2% Tween 20 followed by incubation with secondary antibodies (Supplementary Table 3) diluted 1:10,000 in blocking buffer. The membranes were scanned with Odyssey CLX (LI-COR Biosciences). Semi-quantification of the observed band was performed with Fiji 1.53. β-TUBULIN signal was used to normalize for the input amount.

## Immunohistochemistry

Individual ROs were fixed with 2% paraformaldehyde (PFA) for 15 min at 4 °C, followed by cryoprotection in a sucrose gradient (7.5% 30 min, 15% 30 min and 30% 2 h) at 4 °C prior to embedding in optimal cutting temperature (OCT) compound (Tissue-Tek). Cryosections at 7 µm thickness were rehydrated with PBS for 10 min followed by blocking and permeabilization for 1 h in a buffer containing 10% normal goat serum (Thermo Fisher Scientific) or normal donkey serum (Jackson Immuno Research), 1% bovine serum albumin (BSA), 0.5% Triton X-100. The sections were subsequently incubated in primary antibodies diluted in a buffer containing 3% normal goat serum or normal donkey serum, 1% BSA, and 0.5% Triton X-100 at 4 °C overnight. Unbound antibodies were washed away 3 × 15 min with PBS + 0.1% Tween 20. To allow fluorescence detection, sections were incubated with secondary antibodies and DAPI for 1 h at RT. The secondary antibodies were diluted in the same buffer as that of the primary antibody. Following 2 × 15 min wash in PBS + 0.1% Tween 20 and 1 × 15 min in PBS, the sections were mounted with Prolong Gold (Thermo Fisher Scientific) and imaged with Zeiss Axio Imager. The antibodies used are listed in Supplementary Table 3.

Mouse eyes were collected as previously described with minor modifications[11]. Enucleated mouse eyes were fixed in 4% PFA for 10 min at RT. Following cornea and lens removal, the eyecups were fixed in 2% PFA for 2 h at RT. Cryoprotection by sucrose gradient (20% 1 h, 30% 1 h, 40% 16 h) was carried out at 4 °C followed by embedding in OCT (Tissue-Tek). Cryosections (7 µm) were rehydrated for 10 min in PBS. Permeabilization was performed with 0.1% Tween 20/PBS (Caspase-3 staining) or 0.01% Tween 20/PBS (Glial fibrillary acidic protein (GFAP) staining) for 20 min at RT. Unspecific antibody binding was blocked for 30 min at RT with blocking solution optimized for each staining (GFAP: 0.1% ovalbumin and 0.5% fish gelatine, Caspase-3: 10% normal goat serum and 2% BSA). Primary antibodies were diluted in the corresponding blocking solution and incubated overnight at 4 °C. Unbound antibody was washed 3 × 15 min with PBS and fluorescence detection was enabled by incubation with secondary antibodies and DAPI for 1 h at RT. After washing away unbound antibodies with PBS, the sections were mounted with Prolong Gold (Thermo Fisher Scientific) and imaged with Zeiss Axio Imager. The antibodies used are listed in Supplementary Table 3.

## Off-target analysis

In silico prediction of potential off-targets of candidate AONs were performed using GGGenome (https://gggenome.dbcls.jp/) with the following

settings; (1) both strand, mismatches/gaps and (2) both strand/mismatches. In vitro investigation for potential off-targets were also performed. Briefly, ROs were treated with the lead candidate AON (A7 21-mer) and a sense oligonucleotide (SON) exactly complementary to it, at 10 µM final concentration at day 230 of the differentiation. Ten days post-treatment, the ROs were harvested and pooled (4 ROs per condition). RNA isolation was performed as described above and subsequently submitted for total RNA-seq with rRNA depletion at GenomeScan BV. The demultiplexed fastq files were mapped to human GRCh38.p13 using STAR2 v2.7.10[12] with default settings, followed by feature counting using HTSeq v2.0.2[13]. Cufflinks v2.2.1[14] was used to do FPKM normalization of the expression data. Differential expression analysis was performed using DESeq2 (version 1.39.8)[15]. Gene ontology enrichment analysis was performed using GeneTrail 3.2 (https://genetrail.bioinf.uni-sb.de/start.html), with the following settings; (1) over-representation analysis, (2) all protein coding genes. Differential splicing analysis was carried out by RTC Bioinformatics Radboudumc, utilizing Leafcutter (0.2.9)[16]. The RNA-seq data were deposited in Gene Expression Omnibus with accession number GSE253344.

## PBMC assay

Peripheral blood mononuclear cells (PBMCs) were isolated from healthy donors and subsequently seeded at 500,000 cells/well of 96-well plate in culture medium (Supplementary Table 1). PBMCs were stimulated with AON 7 21-mer at final concentration of 1 and 10 µM for 24 h at 37 °C with 5% CO2. CpG oligonucleotides (10 µM, ODN2395, Invivogen) and R848 (1 µM, TLR7/8 agonist, Invivogen) were included as positive controls. Furthermore, 10 µM of Ultevursen, an AON with 2'MOE/PS chemistry targeting USH2A, which did not show an inflammatory response in a previous study[17], was included as benchmark control. All stimulations were performed in duplicate per donor. At the end of the stimulation period, the supernatant was collected by centrifugation at 300 g for 5 min. The concentrations of proinflammatory cytokines IFNα, IL-6, IP-10, TNFα, MIP-1α and MIP-1β were measured by enzyme-linked immunosorbent assay (ELISA). The kits used are listed in Supplementary Table 3.

## Cytotoxicity and viability assay

Cytotoxicity of the AONs in ROs was determined by a lactate dehydrogenase (LDH) release assay at 24 h and 10 d post-delivery. CyQuant LDH Cytotoxicity assay (Thermo Fisher Scientific C20300) was employed for this purpose, following manufacturer's manual, with the deviation that 1% Triton X-100 (final concentration) was used to lyse RO for max LDH control.

Viability of RO was measured at 10 d post-AON delivery using CellTiter-Glo 3D Cell Viability assay (Promega G9681). A RO in 50 µl medium was lysed with 50 µl CellTiter-Glo reagent, shaking for 5 min. The reaction was incubated for 25 min prior to luminescence measurement using Victor3 Plate Reader.

## AON stability study

The stability of A7 21-mer in vivo was studied in C57BL/6 J mice (Charles River Laboratories). Both male and female mice were used in similar proportion. Intravitreal injection was performed as described previously[18] at 8 weeks of age, in which 10 µg or 50 µg of AON were injected to each eye of the animal. Vehicle (PBS) was included as negative control. Retinas from each eye were collected 1, 2, 4, 6, 8, and 12 weeks post-injection. Since this is a pilot study, there were 3 retinas collected per condition per time-point. The animals were randomised by picking male or female randomly from the cage prior to treatment and thus, allocation in experimental groups. Homogenization buffer (2 mg/ml Proteinase K (Thermo Fisher Scientific), 0.1 M Tris-HCl pH 8.5, 0.2 M NaCl, 0.2% SDS, 5 mM EDTA) was added to each retina to the concentration of 60 mg/ml. The retinas were homogenized using TissueLyserIII (Qiagen) followed by incubation at 50 °C for 4 h. AON concentration in these retina homogenates were determined by probe-based ELISA. Briefly, the plate was coated with 100 nM capture probe (Supplementary Table 2) for 1 h at RT. Excess probe was washed followed by

samples incubation on the coated plate for 1 h at 42 °C. Afterwards, the plate was washed and incubated with 100 nM detection probe (Supplementary Table 2) for 1 h at 42 °C. Anti-Dig-POD was added to detect the Digoxigenin conjugated to the detection probe and incubated at RT for 30 min, followed by addition of TMB substrate for color development. The absorbance was measured at 450 nm. The probe-based ELISA was performed by Ardena BV. The entire study was performed in a blinded setting.

## AON systemic exposure study

To study the possible systemic exposure of intravitreally injected AON in vivo, A7 21-mer was intravitreally injected at 10 µg and 50 µg into the eyes of C57BL/6 J (Charles River Laboratories) mice at 7 weeks of age. Both male and female mice were used in similar proportion. Since this is a pilot study, three mice were included per condition. The animals were randomised by picking male or female randomly from the cage prior to treatment and thus, allocation in experimental groups. Plasma was collected just before injection (0 h) as well as at 2, 6, 24, 48 and 168 h post-injection. Retina, brain, spleen, liver and kidneys were harvested at 168 h post-injection. Homogenization buffer (2 mg/ml Proteinase K (Thermo Fisher Scientific), 0.1 M Tris-HCl pH 8.5, 0.2 M NaCl, 0.2% SDS, 5 mM EDTA) was added to each retina to the concentration of 60 mg/ml. Tissues were homogenized using TissueLyserIII (Qiagen) followed by incubation at 50 °C for 2 h (liver), 4 h (retina, kidney, brain) or overnight (spleen). The AON concentration in these materials was determined by Ardena BV employing probe-based ELISA, similar to the stability study. AON distribution in the retina was visualized by miRNA-scope which is described in detail below. The entire study was performed in a blinded setting.

## miRNAscope

ROs and mouse retinas were fixed and embedded following a similar protocol as described for immunohistochemistry. Probe targeting A7 was designed by ACD Biosystems Bio-Techne (proprietary sequence). The miRNAscope HD (RED) assay (Bio-Techne) was performed on 7 µm cryosections following manufacturer's instructions, with 5 min target retrieval for RO and 15 min target retrieval for mouse retina. Protease reaction was performed for 30 min prior to incubation with A7-specific probe. The slides were counterstained with DAPI and mounted in Prolong Gold (Thermo Fisher Scientific). The sections were imaged with Zeiss Axio Imager.

## Expression of ABCA4 with internal exon 6 deletion

A plasmid containing full-length ABCA4 cDNA was Gateway cloned into pcDNA3-HA/DEST generating an HA-tagged cDNA clone. A 96-bp deletion of exon 6 was introduced by site-directed mutagenesis. The primers used for the site-directed mutagenesis are listed in Supplementary Table 2.

Wild-type plasmid as well as one with the 96-bp exon 6 deletion were transfected into HEK293T cells at 70-80% confluency using 1 µg plasmid and 3 µl Fugene HD (1:3 ratio, Promega). The cells were harvested 48 h post-transfection and subjected to lysate preparation followed by total protein quantification and Western blot.

For the localization study, constructs were transfected into hTERT RPE-1 cells at 70-80% confluency using 1 µg plasmid and 5 µl Fugene HD (1:5 ratio, Promega). Forty-eight hours post-transfection, cells were fixed in 2% PFA/PBS, followed by permeabilization with 1% Triton X-100/PBS. Blocking was performed by incubation in 2% BSA/PBS for 20 min at RT, followed by primary antibody incubation for 1 h at RT. After 3 × 5 min washes with PBS 3 × 5 min, the samples were subjected to secondary antibody incubation for 1 h at RT. The coverslips were washed 3 × 5 min with PBS before mounting using Vectashield with DAPI (Vector Labs). The antibodies used are listed in Supplementary Table 3.

## Statistics and reproducibility

Biological replicates refer to independent AON treatment to PPCs/ROs from separate differentiations or independent transfections from separate passages. Technical replicates refer to separate molecular read-outs from the same sample, e.g. two independent RT-PCRs from one biological replicate. For each experiments described in this study, at least two biological replicates analysed in two technical replicates were included. This is explicitly stated in each figure legend. In all experiments involving AON treatment, significance was calculated by one-way ANOVA followed by post-hoc Dunnett test comparing each AON-treated sample to SON or vehicle (PBS). Pairwise two-tailed t-test was employed to determine significance in experiments comparing (isogenic) control vs patient/mutant.

## Reporting summary

Further information on research design is available in the Nature Portfolio Reporting Summary linked to this article.

# Results

## AON screen in photoreceptor precursor cells and retinal organoids

The noncanonical splice site variant ABCA4 c.768G>T is located at the last nucleotide of exon 6 and reduces the strength of the original splice donor site, favouring the use of an alternative splice donor site 35 nt downstream into intron 6 (Fig. 1a). Functional analysis of this variant in a midigene assay corroborates this prediction by revealing a 35-nt elongation of ABCA4 exon 6[7]. This finding was further validated in patient-derived fibroblasts, in which the 35-nt elongation was also observed after inhibition of nonsense-mediated decay (NMD) (Fig. 1b). NMD inhibition was not necessary to observe this variant in retina-like cells, such as photoreceptor precursor cells and retinal organoids (Fig. 1b).

The high prevalence of the ABCA4 c.768G>T variant, particularly in the North-West of Europe, renders it a highly interesting therapeutic target. To correct the splicing defect, twenty-five 2'MOE/PS AONs were designed in an oligo-walk manner covering almost the entire exon elongation including the new splice site (Fig. 1c, Supplementary Table 4). Photoreceptor precursor cells (PPCs, Supplementary Fig. 1) and retinal organoids (ROs, Supplementary Fig. 2) were generated from a patient-derived iPSC line (Fig. 2a) as well as from its isogenic control. These retina-like models showed consistent endogenous ABCA4 expression and therefore are suitable in vitro models to test the potency of these AONs in modulating ABCA4 splicing. The patient line carries the c.768G>T variant homozygously, allowing evaluation of the AON effect without the influence of another variant of different severity. In the first screening, the AONs were delivered gymnotically into PPCs at 5 µM final concentration for a duration of 10 d. RT-PCR analysis showed a varying degree of correction of the aberrant transcript (Fig. 2b, c, Supplementary Fig. 3a, 4a).

The 10 most potent AONs were selected based on the percentage of wild-type (WT) transcript detected in the semi-quantification as well as the capillary electrophoresis of the PCR product. These 10 AONs were further tested in ROs at 10 µM final concentration for a duration of 10 d. Among the AONs tested, A7 showed the highest %WT transcript with p-value < 0.0001 (Fig. 2d, e, Supplementary Figs. 3b and 4b). Subsequently, this AON was further modified by shortening it from the 3′ end. A 19-mer and a 20-mer version of it were tested along with the original A7 (21-mer) in different concentrations in ROs. Although all AONs were able to rescue the splicing defect, RT-PCR analysis showed that the 21-mer version outperformed the shorter version and it reduced the levels of aberrant transcript in a dose-dependent manner (Fig. 3A, Supplementary Fig. 5a-d).

It is notable that a smaller transcript is observed in the RT-PCR of the RO-derived materials (Fig. 2b), that is more prominent in the ROs compared to the PPCs and whose expression is increased upon AON treatment. Sanger sequencing revealed this transcript to contain a deletion of the last 96 nt of ABCA4 exon 6 (c.673_768del, Supplementary Fig. 6). One of the tested AONs, A1, seemed to completely prevent the expression of both the correct and aberrant transcript, while only expressing the smaller transcript with the partial exon 6 deletion (Fig. 2b). There is a weak splice donor site within ABCA4 exon 6 (Supplementary Fig. 7a), hence the observation of this transcript in untreated patient materials. We hypothesize that blocking of the splice donor site at c.768+35 position activates the splice donor site at

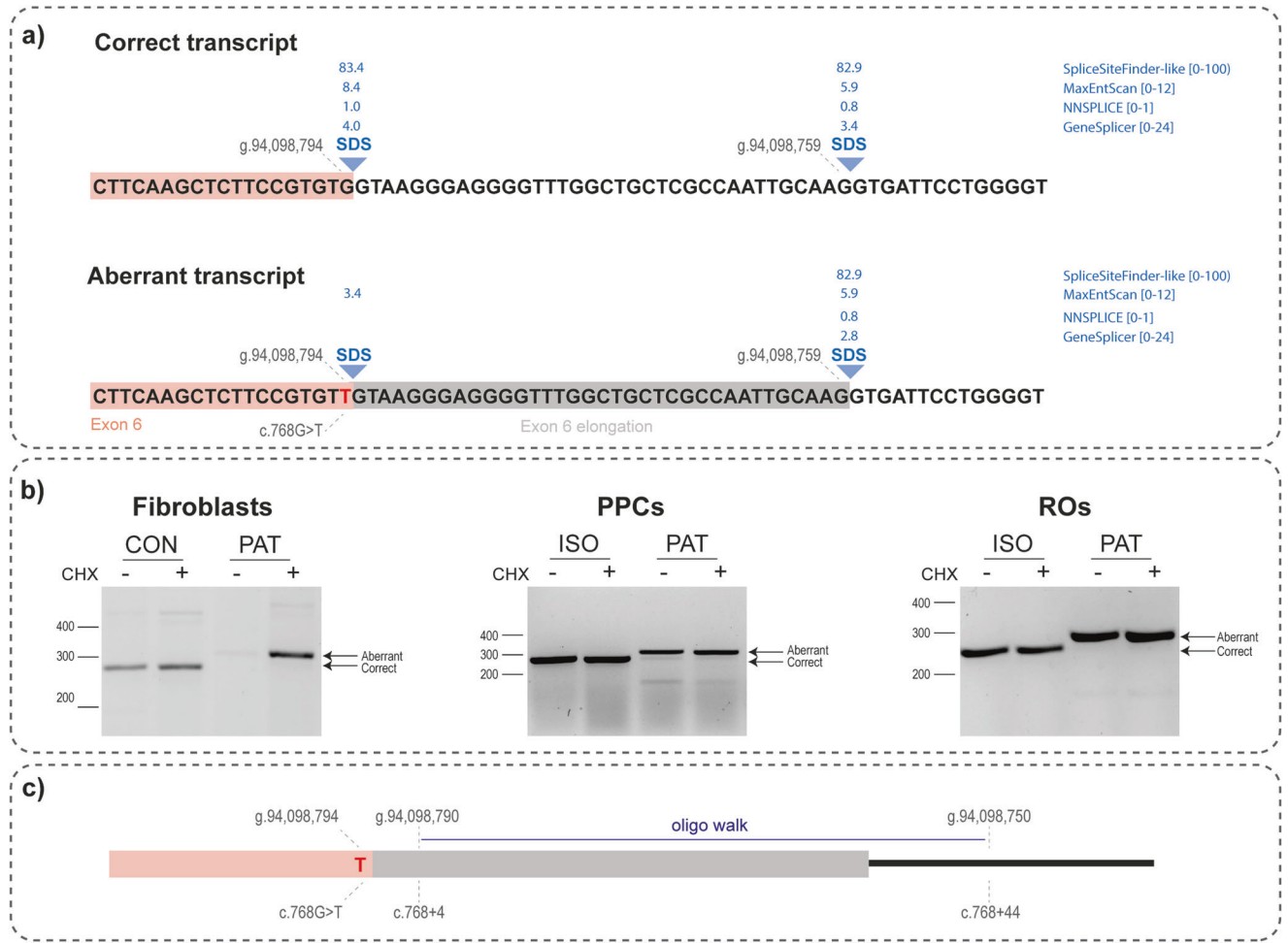

**Fig. 1 | The effect of *ABCA4* c.768G>T variant on splicing. a** A schematic overview of splice donor sites (SDS) of the correct and aberrant transcript. The variant c.768G>T is indicated and the score values assessed by various prediction tools are shown above the SDS. **b** RT-PCR analysis showing the effect of the c.768G>T splicing in fibroblasts, PPCs and ROs. PPCs: photoreceptor precursor cells, ROs: retinal organoids, CON: control individual, PAT: patient-derived material, ISO: isogenic control of patient-derived material, CHX: cycloheximide. **c** A schematic overview of the region included in the oligo walk. The variant c.768G>T is indicated and AON sequences are listed in Supplementary Table 4.

the c.672 position in case the original splice donor site (at position c.768) is also not available. Overexpression of a construct containing this deletion showed ABCA4 expression level in HEK293T cells and a localization pattern in hTERT RPE-1 cells that were similar to the full-length ABCA4 the construct (Supplementary Figs. 7b–e and 8). Despite the similar expression level and pattern, further investigation revealed that the partial exon 6 deletion severely reduced the ATPase activity of ABCA4 (Supplementary Fig. 7f), which needs to be taken into account when estimating the correction potential of candidate AONs (Supplementary Fig. 7g, h).

**Restoration of ABCA4 protein expression**
We hypothesized that the treatment duration needs to be longer to be able to see a correction at the protein level. Therefore, AONs were delivered gymnotically to ROs and harvested at different time-points without additional AON delivery. The treatment duration did not influence the correction efficiency at RNA level (Supplementary Fig. 9a-9b, 10a), whereas 30 d treatment revealed to be the best time-point (Supplementary Fig. S9c-9d, 10b). ROs treated for 30 d with A7 21-mer expressed 50% ABCA4 relative to the isogenic control, in contrast to the null expression in untreated and SON-treated patient ROs (p-value < 0.0001, Fig. 3B, C, Supplementary Fig. 5d, 7h). In line with the observations at the transcript level (Fig. 3A, Supplementary Fig. S5a-c), less ABCA4 expression was observed in ROs treated with A7 19 and 20-mer, compared to the 21-mer version (Fig. 3B, C,

Supplementary Fig. 5d). Immunohistochemical analysis showed that upon 30 d AON treatment, ABCA4 is expressed in the outer segment of the ROs, similar to that observed in isogenic control (Fig. 3D, Supplementary Fig. 11-12). Altogether, our data showed that AONs can rescue the splicing defect caused by *ABCA4* c.768G>T, with A7 21-mer being the most potent among the tested AONs.

**Stability and systemic exposure of the lead candidate AON**
It is very promising to see that with a single AON delivery into ROs, the correction was observed up to 60 d post-delivery which indicates how stable the effect of this AON is in vitro (Supplementary Fig. 9c-9d). To further investigate the stability of this AON, an in situ hybridization assay with a probe specifically targeting A7 was designed and tested. A7 21-mer was well detected in ROs 10 d post-delivery, when it only showed an effect at the transcript level, as well as at 30 d post-delivery, the best time-point at protein level (Fig. 4a). These data provide further evidence on the stability of this AON in vitro. A7 21-mer was detected in all retinal layers of C57BL/6 J mice one week after intravitreal injection with 10 and 50 µg dose (Fig. 4b, Supplementary Fig. 10a). A7 21-mer was also detected in spleen, liver and kidney one week after injection (Supplementary Fig. 13a), although it is cleared from the plasma 6 h after injection (Supplementary Fig. 13b). Despite some variability observed, A7 21-mer was detected in C57BL/6 J mice retina in up to 12 weeks post-injection (Fig. 4c), also illustrating its stability in vivo.

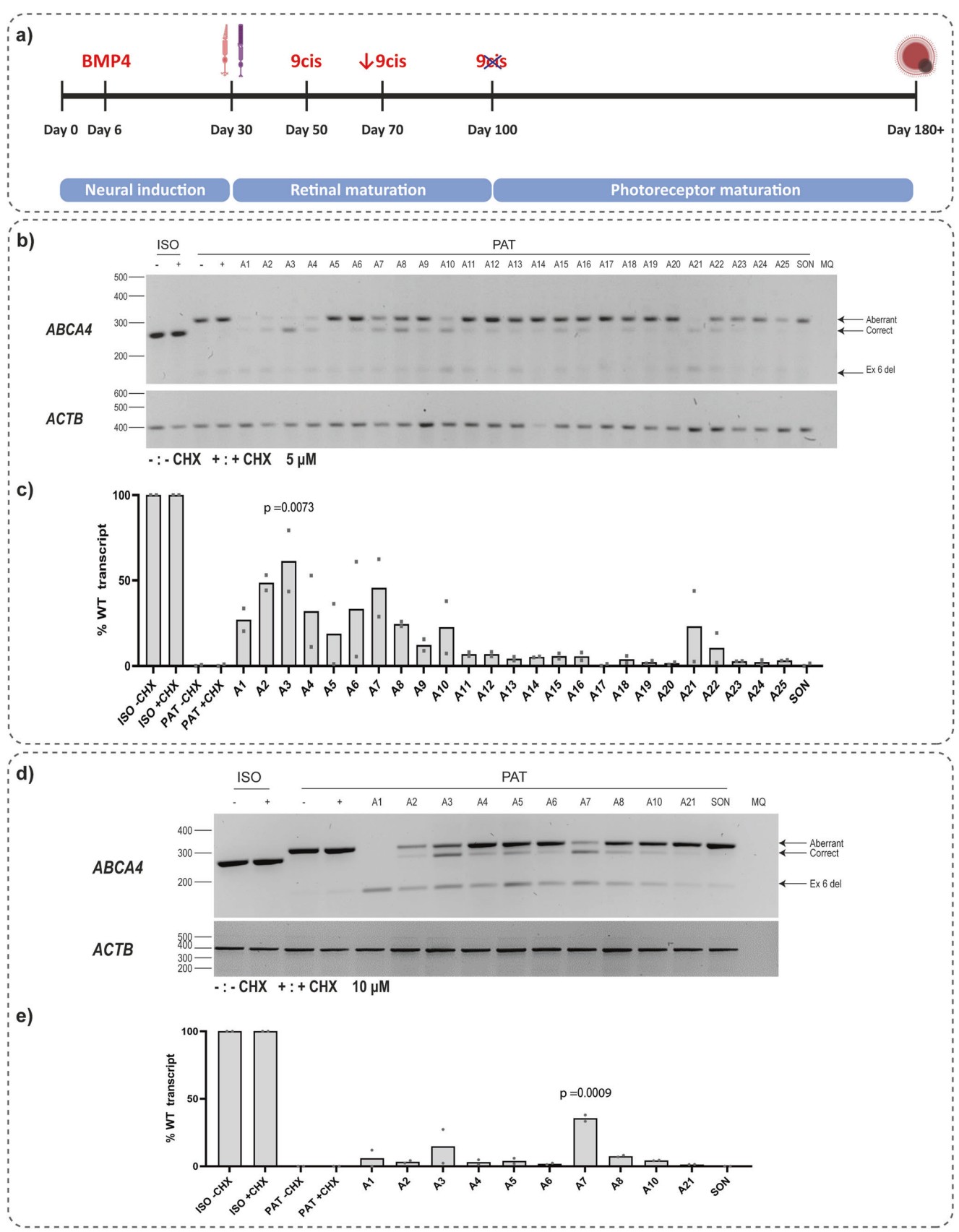

**Fig. 2 | AON screening in PPCs and ROs. a** A schematic overview of the differentiation procedure. Photoreceptor precursor cells (PPCs) are differentiated for 30 d, whereas retinal organoids (ROs) are differentiated for at least 180 d.
**b** Representative RT-PCR analysis of PPCs treated with 2'MOE/PS AONs targeting c.768G>T for 10 d at 5 μM final concentration (*n* = 2). **c** Fiji semi-quantification of WT transcript level based on RT-PCR analysis in (**b**). **d** Representative RT-PCR of ROs treated with 2'MOE/PS AONs targeting c.768G>T for 10 d at 10 μM final concentration (*n* = 2). **e** Fiji semi-quantification of WT transcript level based on RT-PCR analysis in (**d**). **b**, **d** ISO denotes isogenic control and PAT represents patient-derived PPCs/ROs. Three different transcripts were detected, aberrant transcript

containing 35-nt exon 6 elongation, correct/wild-type transcript, and ex 6 del referring to a transcript containing a 96-bp deletion of exon 6 (c.673-c.768). *ACTB* was used as loading control. **c, e** The %WT transcript level illustrated is based on the ratio of aberrant, correct, and partial exon 6 deletion transcripts in each sample, with total signal quantified as 100%. The semi-quantification was performed on at least two PCR products per biological replicate. Each dot denotes the mean of technical replicates of one biological replicate. Significance was calculated by one-way ANOVA followed by post-hoc Dunnett test comparing each AON-treated sample to SON.

## Safety aspects of the lead candidate AON

Besides efficacy, the safety of an AON is obviously of high importance. Total RNA sequencing revealed that the transcriptome profile of ROs treated with A7 21-mer or its sense oligonucleotides (SON 7 21-mer) are very similar (Fig. 5a, Supplementary Fig. 14a). No significantly differentially regulated genes were identified in pairwise comparison between AON-treated vs SON-treated ROs (Fig. 5b, Supplementary Data 1). In contrast, pairwise comparison between isogenic AON-treated ROs vs patient AON-treated ROs and isogenic SON-treated ROs vs patient SON-treated ROs revealed 336 and 395 differentially expressed genes, respectively (Fig. 5b). When comparing these gene lists, 228 genes are identical, comprising genes that are relevant for visual perception and eye structure (Supplementary Fig. 14b, Supplementary Data 2). Genes uniquely found in SON-treated ROs comparisons were related to regulation of wound healing, whereas those found in AON-treated ROs comparisons were too diverse to allow gene ontology enrichment analysis (Supplementary Data 2).

Differential splicing analysis did not reveal any potential alternative splicing events caused by A7 21-mer (Fig. 5c, Supplementary Data 3). A very limited number of significant differential splicing events were observed in the transcriptome data. To study the data further than the p-values, a ranking method based on deltapsi was employed. Deltapsi refers to the splicing difference between the two conditions presented in each pairwise comparison. The aberrant transcript associated with c.768G>T as well as the correct transcript were top of the rank in the comparison between isogenic ROs vs patient-derived ROs (arrows Fig. 5c). The mapped transcriptome data were used to perform manual curation of the top 10 splicing events based on this ranking as well as the significant splicing events. No clear alternative splicing pattern was observed.

Besides transcriptome data, in silico prediction of potential off-targets was also performed using GGGenome (GGGenome | ultrafast DNA search (dbcls.jp)). This analysis revealed limited amount of potential off-target effects (Supplementary Data 4-5) with only 1 mismatch or indel, while it is known that even 1 mismatch already negatively impacts AON binding to its target[19,20]. There was no overlap between the top-ranked differential splicing events observed in the RNA-seq data and the potential off-targets with the least mismatches. Manual curation of the predicted off-target sites also did not show any apparent alternative splicing.

It is also known that phosphorothioate linkage and certain sequence motifs may induce nonspecific pro-inflammatory effects[21–23], whereas ribose modification such as 2'MOE has been shown to reduce risk of inflammatory response[24]. Considering A7 21-mer consists of nucleotides with the 2'MOE modification and a full phosphorothioate linkage, a PBMC stimulation assay was performed to study its immunostimulatory profile. The positive controls, CpG oligonucleotides and R848 (TLR7/8 agonist) as well as the vehicle (PBS) showed the expected stimulation effect. Notably, no significant amount of cytokine and chemokine release were observed upon PBMC stimulation with A7 21-mer (Fig. 6a, S15). With the exception of a few donors, it showed approximately the same immunostimulatory profile to ultevursen, a 2'MOE/PS AON that has been rigorously characterized[25] and has been administered to human subjects.

In vitro safety of A7 21-mer was tested by determining the amount of LDH (lactate dehydrogenase) released into the medium which is a measure of membrane damage and hence, cellular cytotoxicity. A7 21-mer delivery into ROs at final concentration of 10 μM for 24 h and 10 d did not induce

any toxic effect that triggered the release of LDH into the medium (Fig. 6b). This observation is in contrast to the positive control, doxorubicin, a chemotherapeutic agent. Furthermore, the ATP-content measured at day 10 revealed the viability of the AON-treated ROs, whereas doxorubicin-treated ROs were not viable anymore (Supplementary Fig. 16). Furthermore, no Caspase-3 positive cells were detected in mouse retina following intravitreal injection with A7 21-mer (Supplementary Fig. S17). No difference in GFAP activation was observed in retinas of mice injected with A7 21-mer compared to those injected with vehicle (PBS) (Supplementary Fig. 18). Thus, it can be concluded that A7 21-mer does not show any concerning adverse effect in vitro and in vivo, as far as this has been tested.

## Discussion

*ABCA4* c.768 G > T is a prevalent variant underlying Stargardt disease type 1 that leads to a 35-nt elongation of *ABCA4* exon 6, thereby causing a shift in the open reading frame. Twenty-five 2'MOE/PS AONs were designed to rescue the aberrant splicing caused by this variant. Screening of these AONs in PPCs as well as subsequent testing and optimization in ROs allowed us to determine a lead candidate AON, namely A7 21-mer. This particular AON rescued the splicing defect on average 52% at the transcript level, which translated to 50% at the protein level. In vitro and in vivo assessments indicated the stability of this AON without overt negative outcome in the assessments regarding potential off-targets, stability, immunostimulatory profile and cytotoxicity.

Since the exon 6 elongation was only 35 nt, there was limited room to design appropriate AONs to block the recognition of the alternative splice donor site, without interfering with the recognition of the canonical splice site at position c.768. Rescuing such an aberrant splicing has been observed to be challenging[26,27]. Nonetheless, A7 21-mer consistently showed the highest potency in correcting the splicing defect caused by *ABCA4* c.768G>T in a dose-dependent manner, with 10 μM dose as an optimal dose. This is a relatively high dose compared to other AON-based splice correction strategies, particularly those targeting deep-intronic variants[25,28–31]. Besides the commonly observed endosomal escape issue[32], the challenging region may have contributed to the need of such a high dose. Additionally, any modification of the sequence led to substantial reduction in efficacy, indicating the need of a very specific sequence.

An unexpected transcript was detected in the RT-PCR analysis which was more prominent upon AON treatment. Sanger sequencing revealed it to be a partial deletion of *ABCA4* exon 6 (c.673_768del). Since it is an in-frame deletion (96 nt), it is not expected that this transcript will undergo nonsense mediated decay. The partial deletion is affecting amino acids that are located in ECD1 domain, which is predicted to be a flexible region of ABCA4. Studies using wild-type or mutant *ABCA4* cDNA constructs studies showed that the partial deletion did not lead to abnormal expression nor localization, but did disrupt the ATPase activity of the mutant protein. With Western blot analysis or immunocytochemistry in patient-derived retinal organoids, it is impossible to distinguish the wild-type protein from the one lacking 32 amino acids (due to the 96-nt deletion). However, when calculating the percentage of rescue at the protein level, this should be considered.

Although treatment duration did not influence the rescue efficacy at the transcript level, it has a considerable effect at the protein level. Treatment for 30 days resulted in the highest ABCA4 expression that is localized at the

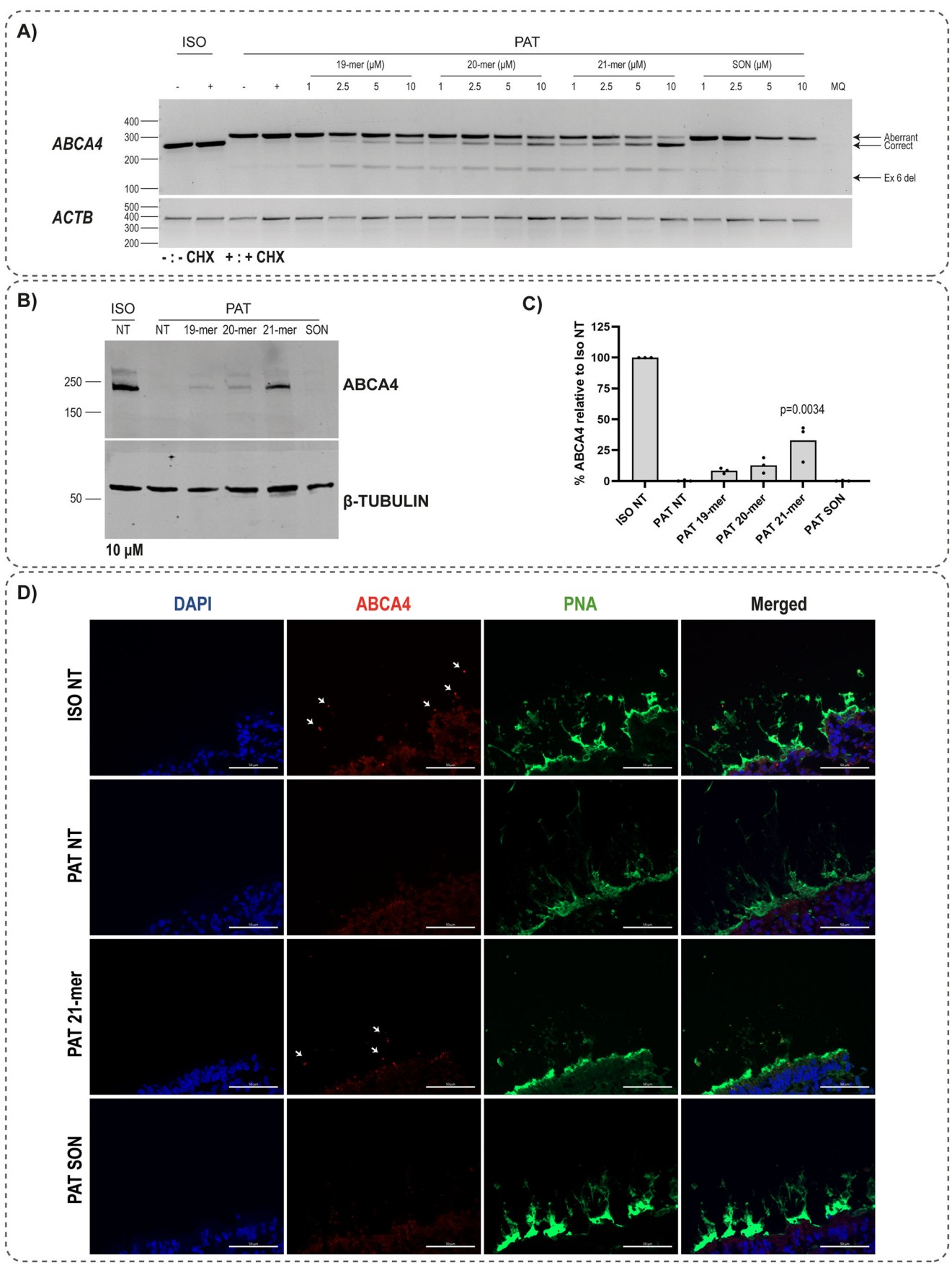

**Fig. 3 | A7 21-mer showed the best potency in rescuing *ABCA4* c.768G>T at RNA and protein level. A** RT-PCR analysis of ROs treated with different lengths and concentrations of A7 for 10 d. ISO denotes isogenic control and PAT represents patient-derived ROs. Three different transcripts were detected, aberrant transcript containing 35-nt exon 6 elongation, correct/wild-type transcript, and ex 6 del referring to a transcript containing the 96-bp deletion of exon 6 (c.673-c.768). *ACTB* was used a loading control. **B** Western blot analysis of ROs treated with A7 19, 20 and 21-mer for 30 d at 10 μM final concentration. **C** Fiji semi-quantification of

ABCA4 expression based on the band at ~250 kDa, normalized to β-TUBULIN expression, relative to Iiogenic control. Each dot denotes the mean of technical replicates of one biological replicate. Significance was calculated by one-way ANOVA followed by post-hoc Dunnett test comparing each AON-treated sample to SON. **D** Immunohistochemal analysis of ABCA4 localization (red, 5B4 antibody) upon 30 d treatment with 10 μM A7 21-mer. PNA (green) is used to mark the outer segments of the ROs, whereas DAPI was used as nuclear marker. Representative images from three biological replicates are shown. Scale bar equals 50 μm.

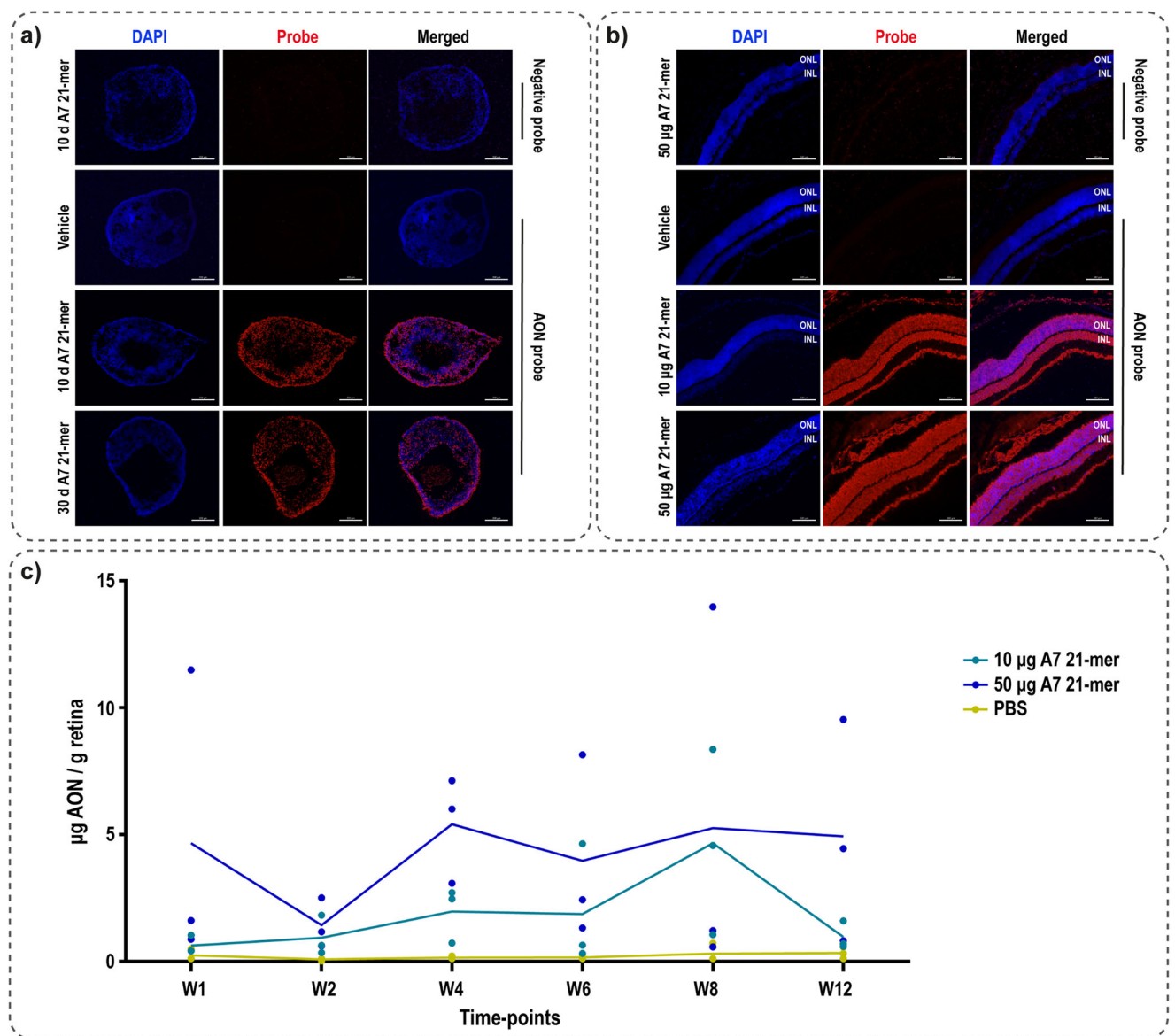

**Fig. 4 | A7 is stable both in vitro and in vivo. a** Detection of A7 21-mer (red) in ROs 10 and 30 d post-delivery by means of in situ hybridization, counterstained with DAPI as nuclear marker. Scale bar equals 50 μm. **b** Detection of A7 21-mer in C57BL/6 J mouse retina 1 week post-injection by means of in situ hybridization.

Scale bar equals 100 μm. ONL: outer nuclear layer, INL: inner nuclear layer. **c** Quantification of A7 21-mer level in C57BL/6 J mouse retina over time (W: week) employing probe-based ELISA. Each dot represents measurement from one retina.

outer segment region of the retinal organoids, in line with previous observations[33,34]. The minimum ABCA4 expression level required for normal vision has not been precisely determined due to the broad spectrum of severity among the variants. Although it has been speculated to be 30–40%[35], a study showed that 10% ABCA4 expression already led to reduced lipofuscin accumulation in *Abca4*[−/−] mice[36]. Considering that treatment with A7 21-mer resulted in on average 50% ABCA4 protein expression, this will likely be sufficient for therapeutic benefit. Additionally, there are STGD1

patients that carry this variant *in trans* with another variant of different severity and hence, do not result in complete absence of functional ABCA4 protein. The residual ABCA4 expression together with the rescued expression may therefore provide enough ABCA4 for normal functioning of photoreceptor cells. The absence of an in vivo model carrying this variant and showing a phenotype (partially) resembling Stargardt disease is an obvious limitation of our approach. Hence, at this stage, the level of ABCA4 protein restoration that has a positive effect phenotypically, cannot be

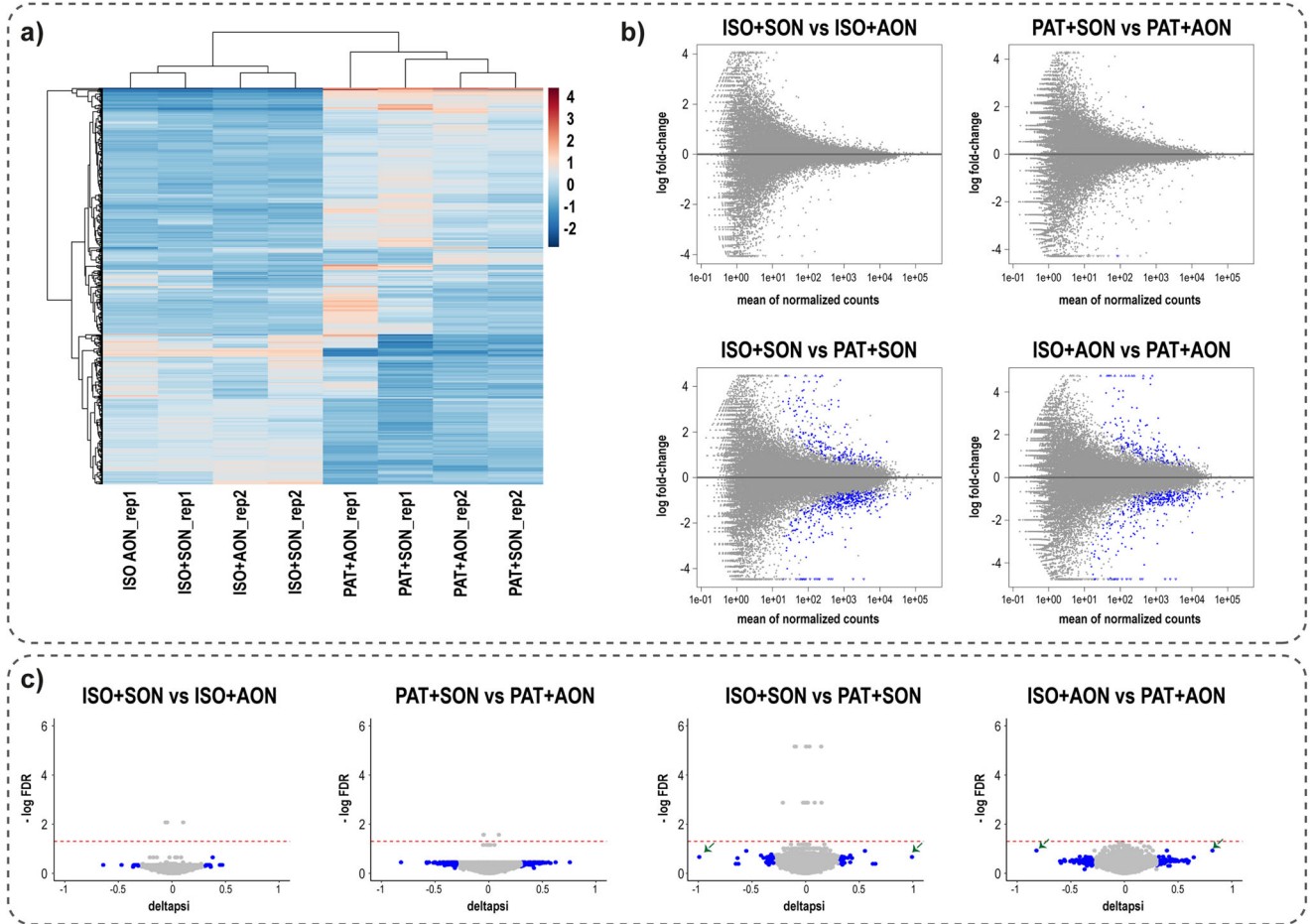

**Fig. 5 | No potential off-targets that cause differential gene expression were identified in total transcriptomics analysis. a** Heatmap of top 500 variable genes comparing all samples included in the analysis. **b** MA-plots of pairwise comparison of differential gene expression. Blue dots represent genes differentially expressed in the pairwise comparison. **c** Volcano plots of pairwise comparison of differential splicing. Blue dots represent genes with deltapsi < −0.3 or >0.3 (data: Supplementary Data 3), red dashed-line denotes FDR threshold of 0.05. Arrows indicate the aberrant transcript associated with *ABCA4* c.768 G > T and the correct transcript. ROs were treated for 10 d prior to analysis. ISO: isogenic control, PAT: patient-derived ROs, AON: A7 21-mer, SON: sense oligonucleotides of A7 21-mer. Data presented in **b**, **c** were generated from two biological replicates shown in (**a**).

assessed. This limitation included in vivo studies testing multiple doses and treatment intervals that could help to define the right treatment regime. Certainly, this dose should also have a balanced safety profile.

AONs with 2′MOE ribose modification, such as used in this study, are known to have enhanced stability[37,38]. We observed rescue of ABCA4 expression up to 60 days post-delivery in vitro, indicating its stability. A7 21-mer is detectable in RO up to 30 days post-delivery and possibly beyond considering its effect is observed up to 60 days. The AON was detected in all retinal layers following intravitreal injection into C57BL/6 J mice and it is detectable in the retina up to the last time-point, 12 weeks post-injection, albeit at a relatively low level compared to the injected amount. Drug elimination from the vitreous occurs either via the anterior or posterior route. The molecules that end up in the blood circulation are then cleared via metabolism in the liver and renal excretion[39]. This is in line with our observation that AON was detected in plasma mainly within a few hours after injection and at later time points in the liver and kidney. A tracer study has shown that a significant amount of [111]In-pentetreotide (1394 Da) was cleared via the anterior route in less than 4 h post-injection[39]. Taking these into consideration, in hindsight, it would have been better to also measure AON level in the retina within hours of injection, to gain insight on the amount of AON that actually reaches the retina.

While intravitreal injection into mouse eyes is commonly performed for efficacy studies, not much is known about intravitreal pharmacokinetics in mouse[39]. AON was detected up to 60 d post-injection in previous mouse studies employing staining-based methods[25,40,41], which allow very sensitive but not quantitative detection. This is demonstrated in this study, i.e., how the strong signal that was detected in in situ hybridization did not correlate to high AON concentration in the quantitative probe-based ELISA. Rabbit is a commonly used species in studying intravitreal drug pharmacokinetics, including AON[25,38,42]. A systematic analysis showed that rabbit and human eye have good correlation in terms of pharmacokinetic parameters[43,44] rendering it to be a good species to perform further in vivo investigation of A7 21-mer.

Safety is a crucial aspect in drug development, which in terms of oligonucleotide therapy can be hybridization-dependent and hybridization independent[45]. No potential off-targets that resulted in gene expression differences were observed, indicating that there is no hybridization-dependent side effect. Our candidate AON has a phosphorothioate (PS) linkage which is known to have potential nonspecific pro-inflammatory effects[21]. Certain palindromic[23] and CpG motifs[22] are known to increase the risk of such inflammatory response, whereas ribose modification, such as 2'MOE has been shown to reduce the risk of inflammatory response[24]. Human primary PBMC stimulation with A7 21-mer did not lead to a significant cytokine or chemokine release. Intravitreal injection into mouse eye with this particular AON also did no lead to extraordinary stress response. Additionally, no cytotoxic effect was observed both in ROs as well as in mouse retina following intravitreal delivery.

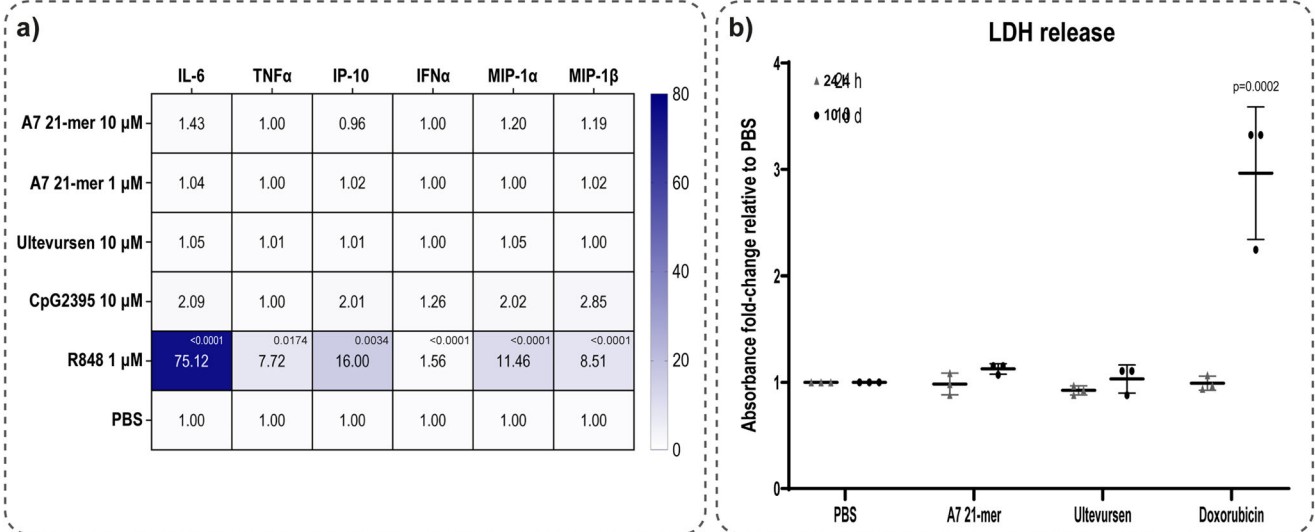

**Fig. 6 | Safety evaluation of A7. a** Heatmap of fold-change cytokines and chemokines released upon 24 h PBMC stimulation with A7 21-mer or controls relative to vehicle (PBS). Geometric mean from 18 healthy donors are presented, with the exception for TNFα (9 donors). **b** Cytotoxicity measured by LDH-release 24 h and 10 d post-delivery into ROs. Three independent experiments including 3 ROs per experimental condition are presented. DOX: doxorubicin, cytotoxic compound used as positive control. For both assessments, significance was calculated by one-way ANOVA followed by post-hoc Dunnett test comparing each AON-treated sample to vehicle (PBS).

In summary, the results obtained in this study serve as a first step towards bringing this therapeutic modality to the patients. It is a comprehensive study comprising a rigorous assessment of the lead candidate AON (A7 21-mer). Follow-up studies (e.g., optimization of the chemistry, in vivo safety and toxicology) should be performed to further evaluate the efficacy and safety of our approach in a relevant animal model, e.g.,. rabbit and/or non-human primate. Finally, although this future therapy is only applicable to a subset of Stargardt disease patients, a substantial number of individuals is expected to benefit from it considering the high prevalence of the *ABCA4* c.768G>T variant.

## Data availability

The source data of Figs. 2c, e, 3C, 4c, 6a and 6b are available in Supplementary Data 6. For Fig. 5, the source data are available in Supplementary Data 1 and 3. The raw RNA-seq data were deposited in Gene Expression Omnibus with accession number GSE253344. Other datasets used and/or analysed during the current study are available from the corresponding author on reasonable request.

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

## Acknowledgements

This study was mainly funded by the Translational Research Acceleration Program of the Foundation Fighting Blindness USA that was awarded to R.W.J.C. and C.B.H. (TA-GT-0521-0799-RAD-TRAP). Also, Retina UK Foundation, grant number GR596 (R.W.J.C), Algemene Nederlandse Vereniging ter Voorkoming van Blindheid, Stichting Blinden-Penning, Landelijke Stichting voor Blinden en Slechtzienden, Stichting Oogfonds Nederland, Stichting Macula Degeneratie Fonds, and Stichting Retina Nederland Fonds (who contributed through UitZicht 2015-31 and 2018-21), together with the Rotterdamse Stichting Blindenbelangen, Stichting Blindenhulp, Stichting tot Verbetering van het Lot der Blinden, Stichting voor Ooglijders, and Stichting Dowilvo supported this study (to A.G. and R.W.J.C.). Initial discovery work was supported by the Foundation Fighting Blindness USA, grant number PPA-0517-0717-RAD (to R.W.J.C. and A.G.). The funding organizations had no role in the design or conduct of this research. They provided unrestricted grants. The authors thank Radboudumc Stem Cell Technology Center, Genomescan BV, Ardena BV and Radboudumc Bioinformatics Technology Center for technical assistance. The authors would also like to thank Prof. Frans Cremers for meaningful discussions and Prof. François Paquet-Durand for providing *rd1* and wild-type retina slides used as positive control in this study.

## Author contributions

D.W.K. and R.W.J.C. were in charge of executing and managing the entire project. D.W.K., A.G., and R.W.J.C. designed the experiments. D.W.K., F.B., and L.D. generated and maintained iPSC-derived retinal models. D.W.K., F.B., L.D., T.Z.T., J.K., A.G., L.L.M., and R.S.M. performed the molecular experiments. A.S. and M.G.N. contributed on the PBMC stimulation assays. S.B. contributed to RNA-seq data analysis. I.V.D. and A.G. contributed to the mouse experiments. A.G., C.B.H., and R.W.J.C. acquired funding. D.W.K. wrote the manuscript. R.W.J.C. supervised the study. All authors read and approved the final manuscript.

## Competing interests

R.W.J.C. is the Chief Executive Officer (interim) of Astherna B.V. C.B.H. is Senior Clinical Advisor of Astherna B.V. F.B. is employed in Astherna B.V. from September 1st 2023. R.W.J.C. and A.G. are inventors on several filed patents describing the use of antisense oligonucleotides (WO2013036105A1, WO2018109011A1, WO2020015959A1, WO2020115106A1, WO2021023863A1) to treat inherited retinal diseases,

including Stargardt disease. These patents have been licensed by Radboudumc to Astherna B.V. M.G.N. is a scientific founder of TTxD, Lemba and Biotrip (not related to the current study).

## Additional information

Consent for publicationNot applicable.

