## [Transparent Peer Review file · Communications Medicine]

Preclinical assessment of splicing modulation therapy for ABCA4 variant c.768G>T in Stargardt disease

Corresponding Author: Professor Rob Collin

Version 0:

Reviewer comments:

Reviewer #1

(Remarks to the Author)

Thank you for the opportunity to review this interesting and well-written article. Its Methods are carefully described and the Figures carefully illustrate their results. The work is an excellent example of the possibility of a Precision Medicine treatment for Stargardt disease. The article provides convincing evidence for the application of an antisense oligonucleotide approach to treat a specific pathogenic variant in the ABCA4 gene. The authors are leaders in this field. The rescue of 50% transcript and protein expression would very likely create a "carrier" state and normalize function. Given their results, I am certain that the article will generate significant interest not only for those with an interest in Stargardt disease, but also groups working on antisense approaches to ocular gene therapy.

Specific comments

1. In the Abstract, prevalence is mentioned and a substantial number of patients. Can they be more specific, or provide estimates? Line 70, pg. 4 says "thousands" – is there a reference? Line 75 says "high prevalence" but provides no value.
2. In the Abstract, the authors could be more specific in providing the length of the "lead" ASO, as in 21-mer, and designation, e.g. A7
3. Please define Eurogentec in the Methods (line 144).
4. Line 548: Whereas additional studies may be conducted in rabbits, a regulator is likely to request a study of toxicity and biodistribution (specifically the retina) after intravitreal injection in a larger animal than the rabbit, with an eye that is closer in size to the human such as the mini-pig.

Minor comments

5. In many instances, pronouns e.g. "a" or "the" are missing or used where not needed. E.g. Line 345 would read better as "... from a patient-derived iPSC line ...". E.g. Line 314 could read better as: "For the localization study, constructs..."
6. Line 304: I would recommend using Prof. Robert Molday without the dr. and include his institution.
7. Line 395: the authors could include "...this AON is in vitro..."
8. Line 413: Prefer: "In contrast, ..."
9. Line 435: Should the authors add "...off-target effects...?"
10. Line 460: Use mice vs. mouse
11. Line 469 would read better as "...splicing defect by 52% at the transcript level which translated to 50% at ..."
12. Line 492 is awkwardly phrased: "Additionally, patients may carry this variant in trans with another variant of different severity and hence, do not result in null expression." Why "may"?
13. Line 541: "...did not lead to an ..."

Reviewer #2

(Remarks to the Author)

This manuscript describes the development of splice-modulating ASOs that address expression of an ABCA4 variant involved in Stargardt disease. The approach is logical, the data are clear and transparent, and the text is well written. My main concern is that the work seems more preliminary than the authors suggest - I would not want a compound at this stage to enter IND enabling trials.

- 1) Even the best compounds seem to produce mediocre splice switching/correct protein production. The authors have not test many compounds. I have no objection to that, as long as the authors do not suggest that they have identified the strongest reasonable candidate for a clinical study. What about other chemistries, maybe mixing in a few LNAs? What about increasing the number of related compounds tested? Either more compounds should be tested, or the conclusions reined in.

2) The problem with the off-target/toxicity analysis is that we don't know the effective dose in vivo. That's a big problem with moving ahead with clinical trials. I understand if an animal model for this mutation is not available, and its absence is no reason to not make these results available. Just tone down the claims.

3) Similar to 1) and 2), the distribution data in Figure 4 is "OK" but is much less impressive than target engagement data would be. The lack of in vivo target engagement data should be noted.

4) Maybe I am mistaken, but in a clinical trial I doubt that target engagement can be monitored. That makes it all the more critical to define target engagement in vivo in animal models prior to moving towards the clinic. Again, I do not ask that this be done now, merely that it be mentioned as being on the "to do" list.

In summary, this is a good paper. Experiments, as far as they go, are rigorously done. It should be amended by making it clear that there are experiments that, in an ideal world of mouse models, should be done prior to clinical trials.

Reviewer #3

(Remarks to the Author)

This study addresses a therapeutic challenge in Stargardt disease type 1 (STGD1), an inherited retinal disease caused by bi-allelic variants in the ABCA4 gene. The study focuses on a recurrent variant, c.768G>T, at the exon-intron junction of exon 6, which leads to a 35-nt elongation causing frameshift and loss-of-function due to aberrant splicing. The study developed 25 2'-O-methoxyethyl antisense oligonucleotides (AONs) to correct this splicing defect. The efficacies of AONs were screened in patient-derived photoreceptor precursor cells and retinal organoids, leading to the selection of a lead AON that restored on average 52% and 50% of wild-type ABCA4 transcript and protein expression, respectively.

The lead AON demonstrated sustained stability in vitro and in vivo without significant safety concerns regarding off-target effects, immune-stimulation, or toxicity. These results showed that the 21-mer version of the AON was particularly effective, exhibiting dose-dependent correction of the aberrant transcript and partial restoration of protein expression.

1. The Materials and Methods section lacks details regarding animal studies such as euthanasia and intravitreal injection. An additional subsection will be required to describe these animal studies.

2. The Materials and Methods section lacks details on how frequently the patient-derived retinal organoids were treated with the lead AON for long-term experiments. If the long-term experiment was performed with a single delivery, please clearly state it.

3. Although the authors performed RT-PCR analysis to screen the 25 AONs, it is crucial to include sequencing data to compare the amounts of aberrant, correct, and ex6 del transcripts.

4. The authors demonstrated that overexpression of a construct containing an exon 6 deletion led to a similar expression level of ABCA4 in HEK293T cells and a localization pattern in hTERT-RPE-1 cells compared to the full-length ABCA4 construct. However, it was not shown whether the construct containing exon 6 deletion did not cause cytotoxicity, with experimental data. Additionally, the authors describe that the partial deletion will not entirely abolish its activity and noted personal communication with Professor Robert Molday. However, the authors should include a functional activity assay.

5. The authors used patient-derived retinal organoids to demonstrate ABCA4 protein amounts following the lead AON treatment. Please describe the protein amount of ABCA4 with exon 6 deletion.

6. It is difficult to appreciate the outer segment of the retinal organoids in the immunohistochemistry data, although the author provided PNA staining. The retinal organoids do not present the architecture of photoreceptors shown in other retinal organoids in the literature. Therefore, more rigorous characterization of the retinal organoids should be performed experimentally.

7. The macular region in STGD1 in humans develops reduced central cone function and degeneration. Additionally, the patient manifests with a loss of central vision due to the degeneration of cone photoreceptors. After further characterization of the retinal organoids, it is recommended that the authors use cone and rod photoreceptor markers for immunohistochemical analysis to visualize the relative expression of ABCA4 in both cone and rod photoreceptors following the lead AON treatment.

8. The signals from ABCA4 in immunohistochemical analysis are very dim. It is highly recommended to improve the signal if this is feasible.

9. The authors briefly described R848 in the Materials and Methods section. Readers will appreciate it if the author describes it in the Results section as well.

10. Although the authors isolated PBMCs, it is highly recommended that authors further isolate NK cells and B cells and repeat the immune-stimulation experiments for better resolution.

11. It has been described that gene therapeutic viruses or siRNA injection into the vitreous can cause vitreous inflammation. Can the authors comment on this in relation to the lead AON?

12. In the Discussion section, please describe how much transcript correction will be required to achieve phenotypic rescue in this particular genotype.

Version 1:

Reviewer comments:

Reviewer #1

(Remarks to the Author)

Dear Colleagues,

Thank you for allowing me to read the revisions with tracked changes. I have no further comments. The authors have successfully addressed all of my questions.

Reviewer #2

(Remarks to the Author)

This was a disappointing response. It lacks thoughtfulness. The authors have done a great job identifying a lead compound, but they seem to continue to be proceeding on the assumption that it is the drug candidate. Mouse or rat toxicology studies are one thing. While they don't mention it, non human primate studies are an obvious next step and those are much more expensive. It would be a shame to see resources poured into investigating a relatively unimpressive lead compound when a better compound might have been discovered through additional screening and rational design one additional modified ASOs. I am not at all convincing that a relatively small expenditure of time and resources cannot identify a better drug candidate. From the standpoint of publication, the manuscript creates the impression that it is relatively simple to move quickly from lead candidate to drug. Rapid translation may be necessary for some patients, but in that case it should be recognized that it comes with a price - higher chance of failure and putting patients through unsuccessful clinical trials.

Reviewer #3

(Remarks to the Author)

The authors made improvements to the manuscript. However, their responses did not seem intended to assist the reviewers but rather to bypass their concerns and let the reviewers deduce the changes. This approach appears somewhat arrogant. All reviewers raised a few significant points, and I recommend that the revision include a thoughtful paragraph in the discussion addressing the limitations of the current studies.

Version 2:

Reviewer comments:

Reviewer #2

(Remarks to the Author)

I am concerned that the authors remain convinced that their mediocre lead molecule is the one that will eventually going into expensive preclinical trials rather than being a starting point for testing other chemistries and other sequences, as well as studies to understand why it is mediocre. My concern is that the authors do not waste resources or shortchange patients chasing a mediocre lead merely because it is in hand now. This drug development business is not easy and overly stubborn adherence to initial lead molecules rarely benefits patients.

I hope that the authors will take a step back and realize that they have not communicated well with supportive referees who were only attempting to offer constructive advice. I understand that ASOs are becoming an increasingly successful modality, but one cannot take that success for granted.

With those pieces of advice out of the way, as currently worded, the manuscript is adequate.

Reviewer #3

(Remarks to the Author)

Thank you for changes

We would like to thank the editors and reviewers for their time evaluating our work and we appreciate that this research is considered of interest for the readership of Communications Medicine. Please find below our point-by-point response to the suggestions and comments from the reviewers.

Reviewer #1 (Remarks to the Author):

Thank you for the opportunity to review this interesting and well-written article. Its Methods are carefully described and the Figures carefully illustrate their results. The work is an excellent example of the possibility of a Precision Medicine treatment for Stargardt disease. The article provides convincing evidence for the application of an antisense oligonucleotide approach to treat a specific pathogenic variant in the ABCA4 gene. The authors are leaders in this field. The rescue of 50% transcript and protein expression would very likely create a “carrier” state and normalize function. Given their results, I am certain that the article will generate significant interest not only for those with an interest in Stargardt disease, but also groups working on antisense approaches to ocular gene therapy.

We thank the reviewer for the kind compliment for our work and the constructive feedback. We agree that addressing the comments of the reviewer will improve the manuscript.

Specific comments

1. In the Abstract, prevalence is mentioned and a substantial number of patients. Can they be more specific, or provide estimates? Line 70, pg. 4 says “thousands” – is there a reference? Line 75 says “high prevalence” but provides no value.

The ABCA4 c.768G>T variant has been reported 430 times in LOVD (<https://databases.lovd.nl/shared/genes/ABCA4>). From the 800 STGD1 Dutch patients included in a study performed by Runhart et al. (2022, PMID:34431609), 103 carried this variant, indicating that is among the most common variants in the North-West of Europe. Due to regional differences in allele frequencies, one cannot just extrapolate these numbers to the rest of the Caucasian population, but based on the prevalence of Stargardt disease worldwide, and the allele frequency of the c.768G>T variant in gnomAD (0.0001051 in the European non-Finnish population), it is expected that there will be a few thousands Stargardt cases with c.768G>T.

2. In the Abstract, the authors could be more specific in providing the length of the “lead” ASO, as in 21-mer, and designation, e.g. A7

This has now been specified in line 42 of the revised manuscript (file with tracked changes).

3. Please define Eurogentec in the Methods (line 144).

Kaneka Eurogentec is a biotechnology company providing life science products, including custom synthesis of oligonucleotides. We have added this information in line 168-169 of the revised manuscript (file with tracked changes).

4. Line 548: Whereas additional studies may be conducted in rabbits, a regulator is likely to request a study of toxicity and biodistribution (specifically the retina) after intravitreal injection in a larger animal than the rabbit, with an eye that is closer in size to the human such as the mini-pig.

Indeed, it is possible that regulatory may request a safety and toxicology studies in other species closer to human, such as non-human primates. Nonetheless, recent advice from our local regulatory body for this specific case is that a safety and toxicology study in rabbit may suffice, taking into account that the 2'MOE/PS chemistry has been tested in humans in the last years, as well as that an in vivo efficacy study cannot be performed in any other species than human due to sequence constraints. Nevertheless, we have adjusted the phrasing in our revised manuscript (line 665 in the file with tracked changes)

Minor comments

5. In many instances, pronouns e.g. “a” or “the” are missing or used where not needed. E.g. Line 345 would read better as “... from a patient-derived iPSC line ...”. E.g. Line 314 could read better as: “For the localization study, constructs...”

We have adjusted these in the revised manuscript.

6. Line 304: I would recommend using Prof. Robert Molday without the dr. and include his institution.

Since we included prof. Molday as a co-author as he performed a functional study, we adjusted line 304 and removed this information (line 341 in the revised file with tracked changes).

7. Line 395: the authors could include “...this AON is in vitro...”

We have adjusted this in the revised manuscript (line 448 in the file with tracked changes).

8. Line 413: Prefer: “In contrast, ...”

We have adjusted this in the revised manuscript (line 472 in the file with tracked changes).

9. Line 435: Should the authors add “..off-target effects..”?

We have adjusted this in the revised manuscript (line 503 in the file with tracked changes).

10. Line 460: Use mice vs. mouse

We have adjusted this in the revised manuscript (line 474 in the file with tracked changes).

11. Line 469 would read better as "...splicing defect by 52% at the transcript level which translated to 50% at ..."

We have adjusted this in the revised manuscript (line 534 in the file with tracked changes).

12. Line 492 is awkwardly phrased: "Additionally, patients may carry this variant in trans with another variant of different severity and hence, do not result in null expression." Why "may"?

We agree that the word 'may' is not correctly phrased here, and adjusted the phrasing of this sentence (line 586-588 in the file with tracked changes).

13. Line 541: "...did not lead to an ..."

With the new findings, we have rephrased this paragraph (line 566-576 in the file with tracked changes).

Reviewer #2 (Remarks to the Author):

This manuscript describes the development of splice-modulating ASOs that address expression of an ABCA4 variant involved in Stargardt disease. The approach is logical, the data are clear and transparent, and the text is well written. My main concern is that the work seems more preliminary than the authors suggest - I would not want a compound at this stage to enter IND enabling trials. *We thank the reviewer for the kind compliment for our work and the constructive feedbacks. We agree that addressing the comments of the reviewer will improve the manuscript. We have toned down the claims made in this manuscript.*

1. Even the best compounds seem to produce mediocre splice switching/correct protein production. The authors have not test many compounds. I have no objection to that, as long as the authors do not suggest that they have identified the strongest reasonable candidate for a clinical study. What about other chemistries, maybe mixing in a few LNAs? What about increasing the number of related compounds tested? Either more compounds should be tested, or the conclusions reined in.

As suggested, we have adjusted the conclusion of our study.

2. The problem with the off-target/toxicity analysis is that we don't know the effective dose in vivo. That's a big problem with moving ahead with clinical trials. I understand if an animal model for this mutation is not available, and its absence is no reason to no make these results available. Just tone down the claims.

We agree with the reviewer that it is hard to predict the effective dose in vivo in the absence of a suitable animal model. As proposed, we therefore toned down our conclusion. Of note, in studies for Sepofarsen (the first splice-switching AON tested in humans for an inherited retinal disease), the dose-range was calculated by extrapolating the volume of an optic cup to the volume of the vitreous humor, so at least there seems to be some correlation possible.

3. Similar to 1) and 2), the distribution data in Figure 4 is "OK" but is much less impressive than target engagement data would be. The lack of in vivo target engagement data should be noted.

We have adjusted the claims of our study and added a statement about the absence of an in vivo model carrying this variant.

4. Maybe I am mistaken, but in a clinical trial I doubt that target engagement can be monitored. That makes it all the more critical to define target engagement in vivo in animal models prior to moving towards the clinic. Again, I do not ask that this be done now, merely that it be mentioned as being on the "to do" list.

Again, the reviewer is right that in an ideal world, we could measure target engagement in vivo but since there is no model available, this is not feasible. As proposed, we rephrased our conclusions and added this to the list of future experiments.

5. In summary, this is a good paper. Experiments, as far as they go, are rigorously done. It should be amended by making it clear that there are experiments that, in an ideal world of mouse models, should be done prior to clinical trials.

We are pleased with the overall appreciation of our work, and agree with reviewer #2 that there is some work to be done before initiating clinical trials. We trust that we have made this more clear in the revised version of our manuscript.

Reviewer #3 (Remarks to the Author):

This study addresses a therapeutic challenge in Stargardt disease type 1 (STGD1), an inherited retinal disease caused by bi-allelic variants in the ABCA4 gene. The study focuses on a recurrent variant, c.768G>T, at the exon-intron junction of exon 6, which leads to a 35-nt elongation causing frameshift and loss-of-function due to aberrant splicing. The study developed 25 2'-O-methoxyethyl antisense oligonucleotides (AONs) to correct this splicing defect. The efficacies of AONs were screened in patient-derived photoreceptor precursor cells and retinal organoids, leading to the selection of a lead AON that restored on average 52% and 50% of wild-type ABCA4 transcript and protein expression, respectively.

The lead AON demonstrated sustained stability in vitro and in vivo without significant safety concerns regarding off-target effects, immune-stimulation, or toxicity. These results showed that the 21-mer version of the AON was particularly effective, exhibiting dose-dependent correction of the aberrant transcript and partial restoration of protein expression.

We thank the reviewer for the constructive feedback. We agree that addressing the comments of the reviewer will improve the manuscript.

1. The Materials and Methods section lacks details regarding animal studies such as euthanasia and intravitreal injection. An additional subsection will be required to describe these animal studies.

The materials and methods required for intravitreal injection and subsequent sacrifice are described in detail in Garanto et al (2022) Delivery of Antisense Oligonucleotides to the Mouse Retina, which is referred to in the Materials and Methods of this manuscript (ref. #18).

2. The Materials and Methods section lacks details on how frequently the patient-derived retinal organoids were treated with the lead AON for long-term experiments. If the long-term experiment was performed with a single delivery, please clearly state it.

All AON deliveries performed in this study were single delivery, including for the long-term experiments. We have now stated this explicitly in the Materials and Methods (line 176-177 in the file with tracked changes).

3. Although the authors performed RT-PCR analysis to screen the 25 AONs, it is crucial to include sequencing data to compare the amounts of aberrant, correct, and ex6 del transcripts.

All the bands observed in the RT-PCR analysis have been Sanger sequenced to validate the transcript size and sequence (a representative is now included in Figure S3). RT-PCR analysis products were subjected to agarose gel electrophoresis followed by semi-quantification using Fiji 1.53 as well as quantification by

automated electrophoresis using Agilent Tapestation. Sequencing data will not add to the comparison of aberrant, correct and ex6 del transcripts level. Although we do have total RNA-seq data, the level of ex 6 del transcript is so low as can be seen in the presented RT-PCR analysis, that it is not possible to accurately quantify it based on those data.

4. The authors demonstrated that overexpression of a construct containing an exon 6 deletion led to a similar expression level of ABCA4 in HEK293T cells and a localization pattern in hTERT-RPE-1 cells compared to the full-length ABCA4 construct. However, it was not shown whether the construct containing exon 6 deletion did not cause cytotoxicity, with experimental data. Additionally, the authors describe that the partial deletion will not entirely abolish its activity and noted personal communication with Professor Robert Molday. However, the authors should include a functional activity assay.

We have performed the ATPase activity assay and added the outcome in Figure S6. With regards to cytotoxicity of the partial exon 6 deletion, it was not observed, taking into account that cells transfected with a construct containing this deletion did not induce cell death and could be used for Western blot, immunocytochemistry and ATPase assay. Furthermore, retinal organoids treated with AON 7 21-mer for 10 days did not show any cytotoxic effect, despite the fact that within this time-frame, the transcript with the partial exon 6 deletion likely would also be translated to protein.

5. The authors used patient-derived retinal organoids to demonstrate ABCA4 protein amounts following the lead AON treatment. Please describe the protein amount of ABCA4 with exon 6 deletion.

Although we can easily distinguish (and therefore reliably measure) the transcript with the partial exon 6 deletion from the wild-type or the one with the 35-nt extension, at the protein level, this is more difficult. First, the protein that is expected to be translated from the 96-nt del transcript may be less stable, and therefore not or only slightly contribute to the overall ABCA4 protein observed in our Western blot analysis or immunocytochemistry. Second, due to the small difference in size, and the location of the epitope of the antibody, both Western blot analysis and immunocytochemistry cannot distinguish between wild-type protein, and the protein lacking the 32 amino acids due to the 96-nt deletion. This has now been explained in our revised manuscript (lines 566-576 in the file with tracked changes).

6. It is difficult to appreciate the outer segment of the retinal organoids in the immunohistochemistry data, although the author provided PNA staining. The retinal organoids do not present the architecture of photoreceptors shown in other retinal organoids in the literature. Therefore, more rigorous characterization of the retinal organoids should be performed experimentally.

Rigorous characterization of the retinal organoids by phase contrast microscopy, gene expression analysis and IHC is provided in the Figure S2. Due to the delicate nature of outer segments of RO, images may vary depending on the embedding, sectioning and staining techniques. We provide images generated with the best effort using the instruments we have available in our laboratory.

7. The macular region in STGD1 in humans develops reduced central cone function and degeneration. Additionally, the patient manifests with a loss of central vision due to the degeneration of cone photoreceptors. After further characterization of the retinal organoids, it is recommended that the authors use cone and rod photoreceptor markers for immunohistochemical analysis to visualize the relative expression of ABCA4 in both cone and rod photoreceptors following the lead AON treatment.

ABCA4 expression together with rod photoreceptor markers (RHO) is shown in Figure S9. We have added extra representative images for better insight. Unfortunately, despite many attempts, the staining of goat-anti ARR3 (cone photoreceptor marker) with mouse-anti ABCA4 was not successful. The ABCA4 antibodies available requires a very specific staining protocol, which turns out to be incompatible with the goat-anti ARR3. Since other cone markers we have available are raised in mouse, we have to use PNA which marks the outer segments of both cone and photoreceptor markers.

8. The signals from ABCA4 in immunohistochemical analysis are very dim. It is highly recommended to improve the signal if this is feasible.

We have added extra images in Figure S8 and S9 for better insight.

9. The authors briefly described R848 in the Materials and Methods section. Readers will appreciate it if the author describes it in the Results section as well.

We have now added this in the Results section (line 514 in the file with tracked changes).

10. Although the authors isolated PBMCs, it is highly recommended that authors further isolate NK cells and B cells and repeat the immune-stimulation experiments for better resolution.

We respectfully disagree with the reviewer on this point, as it is not our intention to study the response of an AON to individual types of cells, but to get an overall view on the immunestimulatory potential. Especially since in the vitreous, there are also different cell types present. PBMCs contain lymphocytes (T cells, B cells, and NK cells), monocytes and dendritic cells. Since we are looking into a system that can mimic immune reaction towards our lead AON in vivo, PBMCs are better suited for this purpose than a purified culture of NK cells and B cells. Furthermore, PBMC stimulation followed by ELISA is

recommended as predictive safety assay following recent guidelines on antisense oligonucleotides therapeutics (Goyenvalle et al. (2023, PMID: 36579950).

11. It has been described that gene therapeutic viruses or siRNA injection into the vitreous can cause vitreous inflammation. Can the authors comment on this in relation to the lead AON?

Intravitreal injections indeed can elicit some type of inflammatory reaction. Gene therapeutic viruses contain a capsid with foreign proteins that can evoke an unwanted response. The latter is not the case for AONs, that are chemically modified RNA molecules dissolved in a neutral salt solution. In addition, there are already a few splice-switching AONs (Sepofarsen, Utevursen) that have been administered to human subjects following intravitreal injection without causing concerning vitreous inflammation.

12. In the Discussion section, please describe how much transcript correction will be required to achieve phenotypic rescue in this particular genotype.

Thank you for the comment. We have discussed this in our revised manuscript (lines 578-597).

We would like to thank the editors and reviewers for their time evaluating our work and we appreciate that this research is considered of interest for the readership of Communications Medicine. Reviewer #1 was satisfied with our previous revision, whilst reviewers #2 and #3 had some remaining concerns. Below, you will find our response to the comments of these reviewers.

Reviewer #2 (Remarks to the Author):

This was a disappointing response. It lacks thoughtfulness. The authors have done a great job identifying a lead compound, but they seem to continue to be proceeding on the assumption that it is the drug candidate. Mouse or rat toxicology studies are one thing. While they don't mention it, non-human primate studies are an obvious next step and those are much more expensive. It would be a shame to see resources poured into investigating a relatively unimpressive lead compound when a better compound might have been discovered through additional screening and rational design one additional modified ASOs. I am not at all convincing that a relatively small expenditure of time and resources cannot identify a better drug candidate. From the standpoint of publication, the manuscript creates the impression that it is relatively simple to move quickly from lead candidate to drug. Rapid translation may be necessary for some patients, but in that case it should be recognized that it comes with a price - higher chance of failure and putting patients through unsuccessful clinical trials.

Author response: We are sorry to hear that this reviewer perceives our previous rebuttal as a disappointing response lacking thoughtfulness. We totally agree with the reviewer that the AON we have now identified as the most potent one within the studies performed, is definitely not ready for clinical use. Perhaps this is caused by a different idea on the definition 'lead molecule'. We are fully aware of the fact that extensive safety / toxicology, including dose-range finding etc. is still needed before a clinical trial can be prepared and initiated. We have tried to better reflect that in the second revision of our manuscript, by removing 'lead AON' and either replace it with 'lead candidate AON' or a different term. Abstract, Introduction, Results and Discussion section have all been adjusted, hopefully in line with the ideas of this reviewer.

Reviewer #3 (Remarks to the Author):

The authors made improvements to the manuscript. However, their responses did not seem intended to assist the reviewers but rather to bypass their concerns and let the reviewers deduce the changes. This approach appears somewhat arrogant. All reviewers raised a few significant points, and I recommend that the revision include a thoughtful paragraph in the discussion addressing the limitations of the current studies.

Author response: We are glad to hear that this reviewer finds our manuscript to have improved. That said, we appreciate the remaining concerns, never having intended to take an arrogant approach in our previous revision. In line with our answer to reviewer #2, we have attempted to better define the stage where our translational research is at this moment, and not to oversell our approach, or to raise the impression that our candidate AON is ready for clinical testing. Although we already listed some limitations of our approach in the previous version, we have now included an extra section on this aspect in the Discussion.